# Increased Sample Entropy in EEGs During the Functional Rehabilitation of an Injured Brain

**DOI:** 10.3390/e21070698

**Published:** 2019-07-16

**Authors:** Qiqi Cheng, Wenwei Yang, Kezhou Liu, Weijie Zhao, Li Wu, Ling Lei, Tengfei Dong, Na Hou, Fan Yang, Yang Qu, Yong Yang

**Affiliations:** College of Life Information Science & Instrument Engineering, Hangzhou Dianzi University, Hangzhou 310018, China

**Keywords:** brain injury, EEG, SampEn, functional recovery, nerve remodeling

## Abstract

Complex nerve remodeling occurs in the injured brain area during functional rehabilitation after a brain injury; however, its mechanism has not been thoroughly elucidated. Neural remodeling can lead to changes in the electrophysiological activity, which can be detected in an electroencephalogram (EEG). In this paper, we used EEG band energy, approximate entropy (ApEn), sample entropy (SampEn), and Lempel–Ziv complexity (LZC) features to characterize the intrinsic rehabilitation dynamics of the injured brain area, thus providing a means of detecting and exploring the mechanism of neurological remodeling during the recovery process after brain injury. The rats in the injury group (*n* = 12) and sham group (*n* = 12) were used to record the bilateral symmetrical EEG on days 1, 4, and 7 after a unilateral brain injury in awake model rats. The open field test (OFT) experiments were performed in the following three groups: an injury group, a sham group, and a control group (*n* = 10). An analysis of the EEG data using the energy, ApEn, SampEn, and LZC features demonstrated that the increase in SampEn was associated with the functional recovery. After the brain injury, the energy values of the delta1 bands on day 4; the delta2 bands on days 4 and 7; the theta, alpha, and beta bands and the values of ApEn, SampEn, and LZC of the cortical EEG signal on days 1, 4 and 7 were significantly lower in the injured brain area than in the non-injured area. During the process of recovery for the injured brain area, the values of the beta bands, ApEn, and SampEn of the injury group increased significantly, and gradually became equal to the value of the sham group. The improvement in the motor function of the model rats significantly correlated with the increase in SampEn. This study provides a method based on EEG nonlinear features for measuring neural remodeling in injured brain areas during brain function recovery. The results may aid in the study of neural remodeling mechanisms.

## 1. Introduction

Brain injury is one of the main causes of death and disability worldwide [1,2]. Functional rehabilitation occurs to some extent during the recovery period after a brain injury, and the mechanism of rehabilitation has long been a major issue in clinical settings and brain science. The current research indicates that blood vessel and nerve remodeling occur during the brain function recovery period [3]. Many reports have described vascular remodeling, such as through optical coherence tomography (OCT), which has shown morphological changes in different types of blood vessels during photothrombotic occlusion and later recovery [4]. However, because the process of nerve remodeling is very complex, it has been studied in few reports to date.

Because neural remodeling involves structural and functional reconstruction, its mechanism can be studied at different levels, such as at the cell and neural circuit levels [5,6]. Because neural remodeling involves cell proliferation and differentiation, synaptic recurrence and connection, and neural network remodeling, changes in the neurophysiological activity can result [6,7]. The changes in these cells can be detected through immunohistochemistry. However, this method requires brain sectioning and staining, which usually must be performed ex vivo, are invasive, and cannot be dynamically detected [5]. Positron emission tomography and functional magnetic resonance imaging (fMRI) can be used for in vivo detection, although their resolution is insufficient to directly record the structural changes during nerve remodeling [7,8]. Electroencephalogram (EEG) signals, characterized by a higher time resolution, can be used to non-invasively record changes in real time, with continuous tracking and a relatively low cost [9,10,11,12,13]. Therefore, EEG signals can serve as a non-invasive and continuous tracking method for neurological remodeling, and provide a means for exploring the mechanism of neural remodeling in the brain.

Generally, the features of EEG signals would be extracted by linear analysis and nonlinear dynamic analysis methods. Studies have shown that the energy of EEG signals differs among patients with epilepsy at different stages; therefore, the energy values may change during the rehabilitation of brain injury [14]. In this work, the EEG energy values were analyzed in order to demonstrate the changes in activity during the rehabilitation of a brain injury. Furthermore, because the brain is a complex nonlinear system [15], it may be suitable for the analysis of EEG signals through a nonlinear dynamic method. Nonlinear dynamic analysis involves many methods, whereas brain activity is extremely non-stationary. The entropy and complexity can be calculated with a short time series and fast speed, without affecting the accuracy [16,17]. Many studies have used approximate entropy (ApEn), sample entropy (SampEn), and Lempel–Ziv complexity (LZC) to study the EEG signals of traumatic brain injury, cerebral ischemia, Alzheimer’s disease, depression, epilepsy, and other diseases [18,19,20,21]. Therefore, it is advantageous to use entropy and complexity to analyze EEG signals. In this paper, energy, ApEn, SampEn, and LZC were selected to analyze the dynamic changes in EEG signals in rats with brain injury and to explore whether the features of EEG reflect neural remodeling during the recovery period.

## 2. Materials and Methods 

### 2.1. Animal Preparation

All of the experiments were conducted on adult male Sprague-Dawley rats (*n* = 44, weighing 180–200 g) obtained from the Experimental Animal Center of Zhejiang Province. The entire experiment is in line with the requirements of the Zhejiang University Ethics Committee (Approval 12511). The animals were reared in ventilated cages individually, in animal holding rooms with controlled temperature (25 ± 2 °C). The light–dark cycle was kept in a 12 h reverse. Standard rat chow and water were provided ad libitum. After the animals had acclimated to their environment, experiments were carried out. The experiments involved 44 rats, which were divided into the following three groups: an injury group (*n* = 21), a sham group (*n* = 13), and a control group (*n* = 10). For the injury group, the unilateral brain area was injured through photochemistry-induced cerebral ischemia. When we observed the dynamic changes in the recovery process after brain injury in the rats, the electrode caps of six rats fell off, and the reference electrodes of three rats were broken during the period of post-operative signal collection. Those nine rats were excluded from the injury group. In the sham group, the rats were implanted with an electrode while keeping brain tissue intact, and did not receive a photochemistry-induced cerebral ischemia injury. The EEG signals were recorded after surgery, and behavioral tests were conducted. The reference electrode of one rat was broken during the period of post-operative signal collection. So, the rat was excluded from the injury group. In the control group, the rats were not injured and did not have an electrode implanted, and only behavioral tests were conducted. 

### 2.2. Photochemistry-Induced Cerebral Ischemia Model and Electrode Implantation

There are many methods for preparing rat models of brain injury. Among those methods, the photochemistry-induced cerebral ischemia model has many advantages, such as minimal surgical trauma, an optional infarction site, a region not limited to the brain areas supplied by the middle cerebral artery (MCA), a controllable extent of injury, favorable long-term survival, and a similarity to human cerebral thrombosis. [22]. Therefore, the photochemistry-induced cerebral ischemia model was chosen for this work. The rats were anesthetized with sodium pentobarbital (1%, 0.5 mL/100 g) and fixed in a stereotaxic apparatus (C6V06-001; RWD Life Science ltd, China). An incision was made along the midline of the skull, as well as 6 × 1.8 mm^2^ and 6 × 1 mm^2^ craniotomies (with the center at 3 mm posterior and 1.9 mm lateral to the bregma), which were set for an optical window and control window in the photochemistry-induced cerebral ischemia model. A small hole for placing the reference electrodes was drilled into the anterior optical window. Before laser irradiation, Rose Bengal solution (2.5 mg/100g, 7.5 mg/mL) was injected intraperitoneally. The occlusion was induced by focal illumination (1 mm diameter focal spot, 30 m W/mm^2^) with a 532 nm laser (CNI Laser; Changchun, China), which was focused on the blood vessels in the window for 30 min [4]. 

The dura was carefully reserved. Four skull screws were drilled into the skull in order to reinforce the implanted electrodes. Two arrays with four electrodes were implanted symmetrically into the optical window and control window. Then, the two recording electrode arrays and the reference electrode were fixed together with dental cement.

### 2.3. Motor Function Assessing

An open field test (OFT) was used to assess the motor function of the animals, which can detect the voluntary activity and exploration behavior of rats [23]. The rats in the OFT experiment were divided into the following three groups: an injury group (*n* = 12), a sham group (*n* = 12), and a control group (*n* = 10). The rats were placed in the room for more than 10 minutes before the behavioral tests. During the experiment, the rats were placed into the field (48 × 48 × 50 cm) with a square scale of 8 cm. Then the number of squares that the rats walked across and the number of times that the rats stood were recorded for 5 min. If the front paws of a rat crossed a square line, or the rat stood up over two-thirds of their whole-body length, one score for the OFT was recorded. The sum of all of the scores within 5 min was the final score of the behavioral test. The surrounding area was kept quiet during testing, and the field was cleaned after each test. The OFT scores were recorded on the first, fourth, and seventh days after brain injury, in order to assess the motor function in the rats. Similarly, the OFT scores of the rats in the sham group and control group were also recorded.

### 2.4. EEG Recording

The EEG acquisition and recording equipment was from the Shanghai Nuocheng Electric Co., Ltd. Signals were recorded from eight channels, in which four channels were in the non-injured brain region, and four channels were in the injured brain region. The two electrodes implanted into the frontal lobe were used as references. The EEG signal was recorded after the signal remained stable for at least 10 min. The signal was set at a sampling rate of 256 Hz, a bandwidth ranging from 0.5 Hz to 30 Hz, and an electrode impedance of <5 kΩ.

The recording of the EEG signals was performed in the injured group and sham group. The EEG signals were recorded on the first, fourth, and seventh days after surgery. To decrease the influence of the fluctuations in the animal physiological state on the EEG, all of the EEG signals were collected in the afternoon.

### 2.5. EEG Signal Processing

In the processing environment of MATLAB R2016b, we first used EEGLAB toolkits to remove the artifacts from the signals, then the power frequency interference was eliminated using a 50-Hz notch filter [24]. The overall calculation was too large, so four time-windows of 10 s were truncated from the acquired EEG data for the analysis. Different frequency bands were decomposed into delta1 (0.5–2 Hz), delta2 (2–4 Hz), theta (4–8Hz), alpha (8–13 Hz), beta1 (13–20 Hz), and beta2 (20–30 Hz) bands. The collected signals had a total of eight channels, as follows: four channels for the brain-injured area and four channels for the non-injured area. Each channel had a characteristic value. A comparative analysis of the four channel signals in the brain injury area and the four channels in the non-injured area on the contralateral side was performed. 

The EEG signal processing includes the following two aspects: linear analysis and nonlinear dynamic analysis. The processing procedure is shown in Figure 1.

#### 2.5.1. Linear Analysis: Band Energy

The EEG signals have different rhythms in the different brain states. The EEG signal amplitude effectively represents the strength of the signal. The square of the amplitude represents the energy values of the EEG signals. The analysis of the EEG signals is usually based on the amplitude and frequency components. Therefore, the EEG signals were divided into different frequency bands. Wavelet decomposition was used to divide the signal into different frequency bands, and the energy value of wavelet coefficient was calculated. The EEG signal was decomposed into five layers with a db3 wavelet base. The energy value of a signal in a band was represented by the energy value of a certain wavelet coefficient. Then, the energy of each frequency band was calculated according to the following energy formula:(1)$$E_{j,i}=\underset{K}{\sum }{\left(d_{j}^{i}\left(k\right)\right)}^{2}$$
where k is the natural number, and $d_{j}^{i}\left(k\right)$ are wavelet coefficients of the *i*-th node of the *j*-th layer. The mean values of the four time-windows of each frequency band were calculated as the final values of the signals.

#### 2.5.2. Nonlinear Dynamic Analysis: ApEn, SampEn, and LZC

In this study, ApEn [25], SampEn [26], and LZC [27] were used to analyze the dynamic changes in EEG in the rats with a focal brain injury. ApEn can be used to detect the probability of a pattern that has not appeared in EEG signal for a period of time. It is an index reflecting the overall characteristics of the signal from the perspective of measuring the complexity of the time series [28,29]. ApEn (m, r, and N), where m represents the length of the run, r represents the tolerance window, and N is the number of points in the time series. First, we need to give N data points for the time series of {a(n)} = a(1), a(2),......, a(N), then calculate ApEn according to the steps given below [28], as follows:
Define m vectors A(1), A(2), A(3),…, A(N) according to the following
A_m_(i) = [a(i), a(i + 1), a(i + 2), . . . , a(i + m − 1)] for 1 ≤ i ≤ N − m + 1(2)Calculate the maximum norm, which can be obtained from the distance between A_m_(i) and A_m_(y)
(3)$$d[A_{m}(i),A_{m}(y)]=\max\left|a\left(i+t\right)-a\left(y+t\right)\right|\;\mathrm{for}\;1\leq \;t\;\leq \;m-1;\;1\;\leq \;i,\;y\;\leq \;N-m+1,i\;\neq \;y$$Calculate the number of y (y = 1 ... N − m + 1, y ≠ *i*) according to the known A(i), provided that d [A(i), A(y)] ≤ r is expressed as M^m^ (i). Then, for *i = N − m*
(4)$$D_{i}^{m}\left(r\right)=\frac{M^{m}\left(i\right)}{N-m+1}\;,\;\mathrm{for}\;y\;=\;1\;\ldots \;N\;-\;m\;+\;1$$Calculate the natural logarithm of each $D_{r}^{m}\left(i\right)$ and its average
(5)$$\Psi^{m}(r)\;=\;\frac{1}{N-m+1\;}\sum_{i=1}^{N-m+1}lnD_{i}^{m}\left(r\right)$$Get $D_{i}^{m+1}$(e) and $\Psi_{}^{m+1}$*(e)* through repeat step (1) to step (4), incrementing the dimension by one each timeFinally, ApEn is calculated according the following
ApEn(m,r, N) = Ψ^m^(r) − Ψ^m+1^(r)(6)

In our study, both ApEn and SampEn were calculated using m = 2 and r = 0.2.

SampEn was used to measure the order of a time series. The larger the value of SampEn, the higher the complexity of the corresponding EEG signal. The first three steps of the method of calculating SampEn and ApEn are the same, the different steps are as follows:
Calculate the natural logarithm of each $D_{i}^{m}\left(r\right)$, and its average
(7)$$B^{m}\left(r\right)\;=\;\frac{1}{N-m\;}\sum_{i=1}^{N-m}{\mathrm{lnD}}_{i}^{m}\left(r\right)$$Increase the dimension to *m* + 1, get $D_{i}^{m}$(r) and $B_{}^{m}$(r) through Formulas (2)–(4) and (7)Finally, SampEn is computed by the following: (8)$$\mathrm{SampEn}\;(m,r,\;N)\;=-\ln[D^{m}\left(r\right)/B^{m}\left(r\right)]$$

LZC indicates the occurrence rate of the new pattern in the time series of an EEG signal, and the value of LZC is proportional to the occurrence rate of the new pattern. [27]. The time series of A (a_l_, a_2_, a_3_, ..., a_n_,) is known. Calculate the average b of A. Then, compare each value of A with b; if it is greater than b, set it to 1, and vice versa to 0. Then, reconstruct a new 0–l string. Assume that the given string (c_1_, c_2_, c_3_, ..., c_n_) has been reconstructed according to the rule. If c_r_ is newly inserted, it is not simply obtained by copying (c_1_, c_2_, c_3_, ..., c_r−1_), but by C = (c_1_, c_2_, c_3_, ..., c_r_, •). The last added point indicates the insertion. You can ask if D = c_r+1_ is included in string C. If true, it can be obtained simply by copying the symbol from C. This is equal to the question of whether DD is included in CDπ, where CDπ denotes a string consisting of C and D, and π denotes that the last digit must be deleted (i.e., CDπ = C).

The mean values of the four 10-s time-windows of each of the features were calculated as the final values of the signals.

### 2.6. Statistical Analysis

The OFT scores, energy of each band, and nonlinear features were analyzed using the Kruskal–Wallis H-test, Wilcoxon signed-rank test, and test of the Kendall coefficient of concordance W in SPSS v.22 software (SPSS Inc., Chicago, IL). All of the results are expressed as median. A Spearman correlation analysis was performed between these features and the OFT scores of the rats after brain injury. *p* < 0.05 was considered statistically significant, *p* < 0.01 was considered highly statistically significant.

## 3. Results

### 3.1. Comparative Analysis of the OFT Scores among the Injured Group, Sham Group, and Control Group

We first compared the OFT results in the sham group, control group, and injured group on days 1, 4, and 7, as shown in Figure 2a. The differences in the OFT scores in the three groups were statistically analyzed with a Kruskal–Wallis H-test, and we performed a pairwise comparison.

As shown in Figure 2a, the differences in the OFT scores of the three groups are significant (*p* < 0.01) on days 1 and 4, but there was no significant difference on day 7. The OFT scores of the injured group are lower than the sham group and control group on days 1, 4, and 7 after brain injury. The differences between the injury group and sham group, and the injury group and control group were significant (*p* < 0.01) on day 1. The difference between the injury and control group was significant (*p* < 0.01) on day 4, but the sham group and the control group were similar on day 4 (*p* > 0.05). The three groups were similar on day 7 (*p* > 0.05). With the recovery process (days 1, 4, and 7) after the brain injury, the OFT results in the injured group gradually approached those of the sham group and control group.

Further investigation of the differences in the OFT scores among the injured group, sham group, and the control group in days 1, 4, and 7 were performed with test of the Kendall coefficient of concordance W, as shown in Figure 2b. The OFT scores of the injured group showed an upward trend in days 1, 4, and 7, and the differences were significant (*p* < 0.01), the sham group showed an upward trend over days 1, 4, and 7, but the differences were not significant (*p* > 0.05), whereas the control group showed minor differences and a fluctuating trend over days 1, 4, and 7.

### 3.2. Comparative Analysis of EEG Signals in the Injured Area and the Symmetrical Non-Injured Area

We then compared the energy of the EEG signals in the injured area and non-injured area on days 1, 4, and 7 in the injury group, as shown in Figure 3a–c. The values of the nonlinear features (ApEn, SampEn, and LZC) in the two areas were compared, as shown in Figure 3d. The differences in the different areas were evaluated with the Wilcoxon signed rank test.

Figure 3a–c showed that the energy values of all of the frequency bands in the non-injured area were higher than those in the injured area on days 1, 4, and 7. The energy values of the theta, alpha, and beta (beta1 and beta2) bands between the injured area and non-injured area were significantly different (theta: *p* < 0.05; alpha and beta: *p* < 0.01) on day 1. The energy values of the delta (delta1 and delta2), theta, alpha, and beta bands were significantly different (delta1: *p* < 0.05; delta2, theta, alpha, and beta: *p* < 0.01) on day 4. The energy values of the delta2, theta, alpha, and beta bands between the injured area and non-injured area were significantly different (theta: *p* < 0.05; delta2, alpha, and beta: *p* < 0.01) on day 7.

Figure 3d showed that the values of ApEn, SampEn, and LZC in the non-injured area were higher than those in the symmetrical injured area on days 1, 4, and 7. There were significant differences (*p* < 0.01) in the ApEn, SampEn, and LZC values in the injured area and non-injured area on days 1, 4, and 7, excluding the LZC values on day 7 (*p* < 0.05). 

### 3.3. Dynamic Change Analysis of EEG Signals in the Injured Area

In the rehabilitation process after a brain injury, what are the trends of the changes in the EEG features in the injured area? Addressing this question is an important way to study the dynamic changes during brain rehabilitation. We investigated the changes in features on days 1, 4, and 7 after brain injury in the brain injured group and sham group, as analyzed with test of Kendall coefficient of concordance W, as shown in Figure 4a–i.

Figure 4a showed the changes in energy values of the delta1 band of the EEG in the injured area on days 1, 4, and 7. These feature values of the injury group fluctuated over time after a brain injury. The energy values of the delta1 on days 1, 4, and 7 of the injury group were lower than the sham group.

Figure 4b–d showed the changes in the energy values of the delta2, theta, and alpha bands of the EEG in the injured area on days 1, 4, and 7. These feature values of the injury group increased with time after brain injury, but the differences were not significant (*p* > 0.05). The energy values of delta1 on days 1, 4, and 7 of the injury group were lower than the sham group, while the values of the delta2 and theta bands of the sham group fluctuated over time after a brain injury, and the values of the alpha band of the sham group increased with time (*p* > 0.05).

Figure 4e,f showed the changes in energy values of beta bands of the EEG in the injured area on days 1, 4, and 7. These feature values of the injury group increased in volatility with time after brain injury, and the differences were significant (*p* < 0.05), while the beta bands values of the sham group fluctuated over time (*p* > 0.05). The energy values of the beta bands of the injury group were lower than the sham group.

Figure 4g,i showed that the values of ApEn and SampEn of the EEG of the injury group increased on days 1, 4, and 7, and the differences were significant (*p* < 0.01), while the SampEn values of the sham group fluctuated over time. The values of ApEn and SampEn of the injury group were lower than the sham group on days 1 and 4, and the value of the injury group on day 7 was approximately the same as the value of the sham group on day 4 and day 7.

Figure 4h showed that the values of LZC increased in the injury group and fluctuated over time in the sham group, but these differences are not significant (*p* > 0.05). 

### 3.4. Relationship between EEG Features and Motor Function Recovery

The EEG complexity measures were previously thought to be associated with changes in brain function in stroke patients. Although the relationship between EEG complexity and behavioral outcomes is not well studied, the energy in various bands, ApEn, SampEn, and LZC, may help predict recovery after brain injury. Therefore, we performed a Spearman correlation analysis of these features with the OFT scores of rats after a brain injury, as shown in Table 1. To show the correlation results in Table 1 more clearly, Figure 5a,b were constructed.

Figure 5a,b showed that the energy values of the delta bands (including the delta1 and delta2 bands) were inversely proportional to the OFT scores, but the results in Table 1 indicate that the correlations were weak. Other features such as the energy values of the theta, alpha, beta1, and beta2 bands, LZC, and ApEn were positively correlated with the OFT scores, but the correlations were not significant (*p* > 0.05). However, SampEn (r = 0.393, *p* < 0.05) was significantly correlated with the OFT scores. 

With the progress of the rehabilitation, the SampEn of the EEG signals of the injured group showed a certain degree of correlation with the OFT scores. This leads to wondering if these results were confounded by the degree of motion artifacts as a result of the improvement in motor capacity. Thus, we also calculated the correlations of SampEn with the OFT scores of the sham group during the progress of the rehabilitation. As shown in Figure 5c, there is no correlation between the SampEn and OFT scores in the sham group (SampEn, r = −0.025; *p* > 0.05).

## 4. Discussion

In this study, we compared the EEG features of a bilateral brain area at different times after a unilateral brain injury. The values of the delta1 band on days 4, the delta2 band on day4 and day 7, theta band, alpha band, and beta bands energy, as well as ApEn, SampEn, and LZC on days 1, 4, and 7 in the injured area were significant lower than those in the non-injured area. Traumatic brain injury studies have shown that brain injury appears to be associated with decreased energy, especially in the alpha band [30]. Liyu Huang studied the ApEn of EEG signals in the acute phase of brain injury caused by unilateral carotid artery ligation, and found lower ApEn values in the cerebral ischemic injured area than the non-ischemic area [31]. On the basis of the EEG analysis method with a symmetric electrode, Li Yi and Zhao Bo quantified the changes in EEG features caused by brain injury by calculating the ratios of the extracted EEG features in the symmetrical brain area, and found that the feature ratios of different brain areas significantly differed [32]. The results of this study were consistent with prior research, to some extent. The differences in energy values between the theta band of the two symmetrical brain areas became smaller during the recovery process. 

To study the changes in EEG features during recovery after a brain injury, we studied the dynamic changes in EEG features in the injured area. The results indicated that the values of the beta bands, ApEn, and SampEn of the EEG signal in the injured area showed an increasing trend during the recovery process (days 1, 4, and 7 after injury). Traumatic brain injury studies have shown that brain injuries appear to be associated with reduced energy. Conversely, recovery from a brain injury is associated with increased energy [30]. The complexity of the EEG signal can be reflected by the entropy value. The more active the brain is, the greater the entropy value is [19,26,28,29]. Because the electrical activity of the cells in the injured brain area is inhibited, and some of the cells may even be apoptotic, the complexity of the injured area is relatively low, and the corresponding entropy values are diminished as well. The brain is reconstructed during the recovery of the injured area (including the proliferation and migration of cells), thus potentially increasing the complexity of the injured area and the corresponding entropy values. These phenomena indicate that changes in the EEG signals during rehabilitation after a brain injury are meaningful for studying the mechanisms of brain rehabilitation. 

In this study, OFT experiments were also conducted so as to evaluate the motor functions of brain-injured rats, control rats, and sham-operation rats. The results of the OFT experiments showed that the motor function scores of the injury group gradually increased during recovery from a brain injury. The scores of the OFT test of the injured group were similar to those of the control group on day 7. The correlations between the motor function and the EEG signal feature values of injured rats were analyzed. The result is that the SampEn was correlated with the motor function scores. In order to exclude speculation about whether these results were confounded by the degree of motion artifacts as a result of the improvement in motor capacity, we also comparatively analyzed the corresponding data of the sham group rats. The results showed that there is no correlation between the SampEn and OFT scores. The results demonstrated that the values of SampEn were correlated with the motor function scores during rehabilitation after brain injury. Chen Shiwen’s research has indicated that motor function changes after brain injury in rats [33]. Abnormal brain function is related to behavioral deficits, and the non-linear complexity of EEG signal can reflect this abnormal brain function. Studies have shown that changes in neuroplasticity are the basis for the recovery of motor function after brain injury [34]. These results suggest that the changes in the EEG features caused by neuronal activity during nerve remodeling in injured brain areas might be a good indicator of brain function recovery.

Neurological repair after cerebral ischemia is closely related to cerebral blood flow. New microvessels attract neurotrophic cells by releasing some growth factors, such as stromal (SDF-1) or (VEGF), thus creating a suitable microenvironment for neurotrophic cell migration, homing, and differentiation [35]. Simultaneously, nerve regeneration enhances the formation of new blood vessels by stimulating the release of VEGF [36]. The synergistic regulation of angiogenesis and nerve regeneration together promote the repair of nerve function during brain function recovery. Shanshan Yang and Kezhou Liu used OCT to show the reconstruction of various types of blood vessels after cerebral ischemia [4]. So, it is very likely that neural remodeling also happens in this process. This study used the same method of cerebral ischemic injury. Our research showed that SampEn in the injured brain area increased gradually, thus suggesting that more neuronal activity patterns might appear, and that the values of SampEn are closely related to neural remodeling in the recovery of brain function. Among the analytical methods, multimodal combination methods, such as MRI combined with EEG to detect the dynamic changes in a brain injury model during the recovery process, may be favorable to study the neural remodeling mechanism after brain injury. 

## 5. Conclusions

By studying the dynamic changes in energy, ApEn, SampEn, and LZC in time and space in cortical EEG signals in both injured and non-injured brain regions in rats with unilateral brain injury, and comparing that with a sham group, we found that the SampEn value of the EEG signal changed during the recovery of rats, which was also accompanied by the recovery of brain function. Therefore, SampEn in the EEG signal can be used as a method for detecting neural remodeling. The functional recovery behavior of neurons is key to the treatment of brain injury. A more comprehensive and in-depth study of the complex mechanisms of cerebral ischemia and nerve injury repair and remodeling should be helpful for the clinical treatment for cerebral ischemic diseases or other diseases involving brain damage.

## Figures and Tables

**Figure 1 entropy-21-00698-f001:**
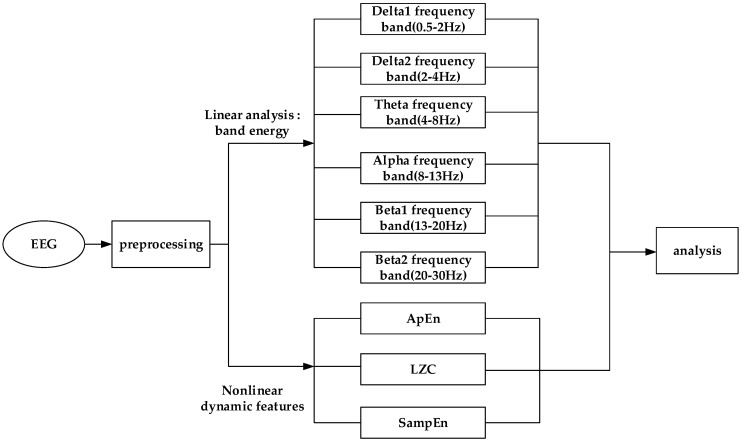
Flow chart of the data processing procedure. EEG—electroencephalogram; LZC—Lempel–Ziv complexity; SampEn—sample entropy; ApEn—approximate entropy.

**Figure 2 entropy-21-00698-f002:**
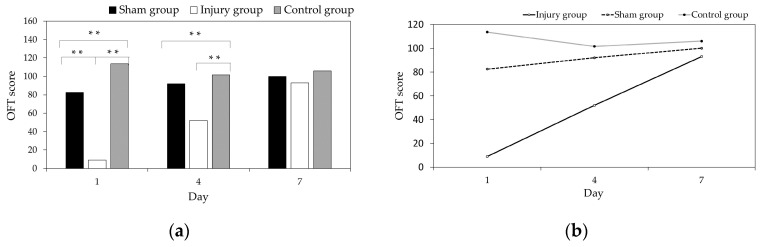
Open field test (OFT) scores in the injured group, sham group, and control group (* *p* < 0.05, ** *p* < 0.01). (**a**) Comparison of differences between groups; (**b**) Comparison of three groups over time.

**Figure 3 entropy-21-00698-f003:**
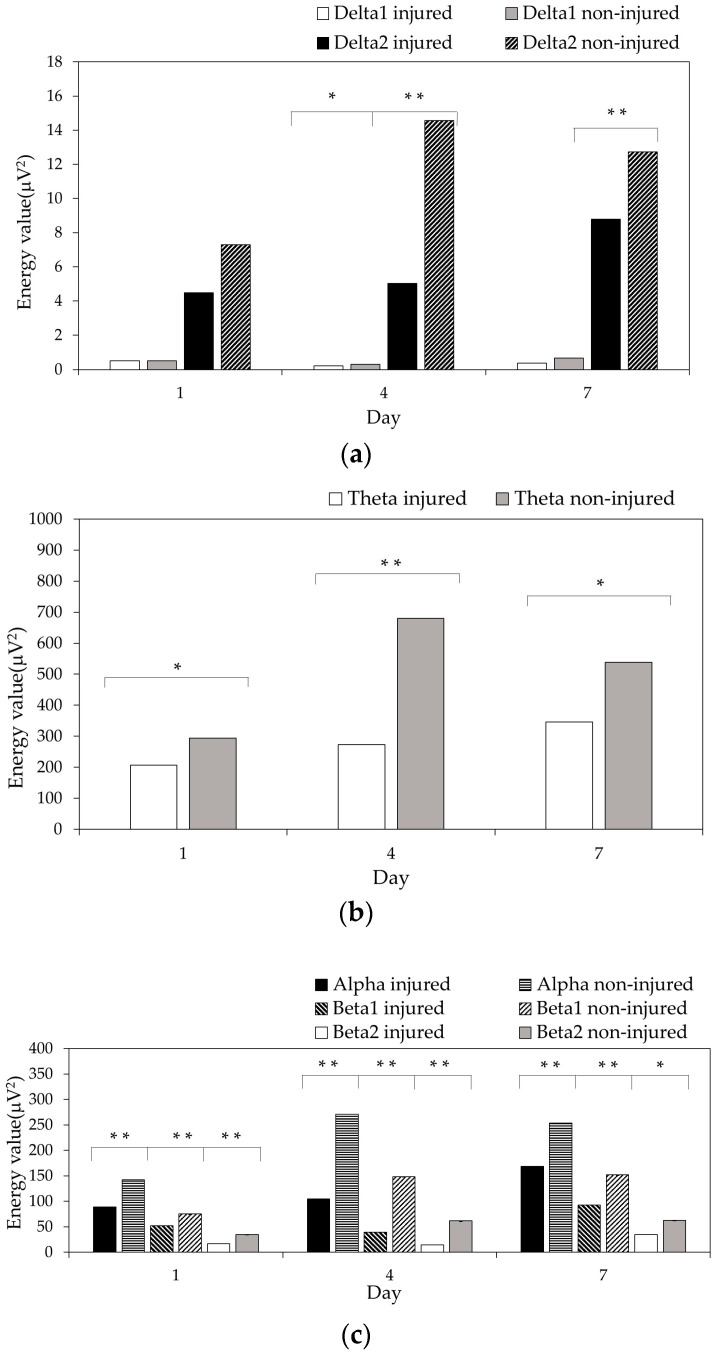
Comparison of EEG signal features between the injured area and symmetrical non-injured area. (**a**) Energy value in the delta band; (**b**) energy value in the theta band; (**c**) energy value in the alpha and beta (beta1 and beta2) bands; (**d**) ApEn, SampEn, and LZC (* *p* < 0.05, ** *p* < 0.01).

**Figure 4 entropy-21-00698-f004:**
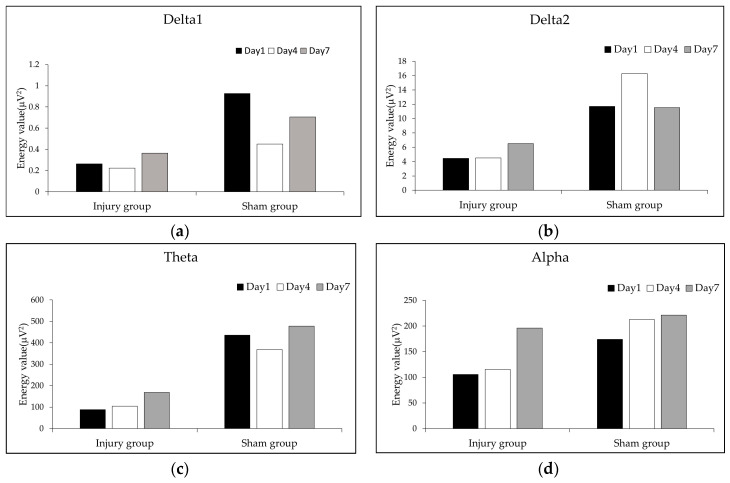
The changes in the EEG feature values in the injured area on days 1, 4, and 7 after brain injury. (**a**) Energy values of the delta1 band; (**b**) energy values of the delta2 band; (**c**) energy values of the theta band; (**d**) energy values of the alpha band; (**e**) energy values of the beta1 band; (**f**) energy values of the beta2 band; (**g**) values of ApEn; (**h**) values of LZC; (**i**) values of SampEn (* *p* < 0.05, ** *p* < 0.01).

**Figure 5 entropy-21-00698-f005:**
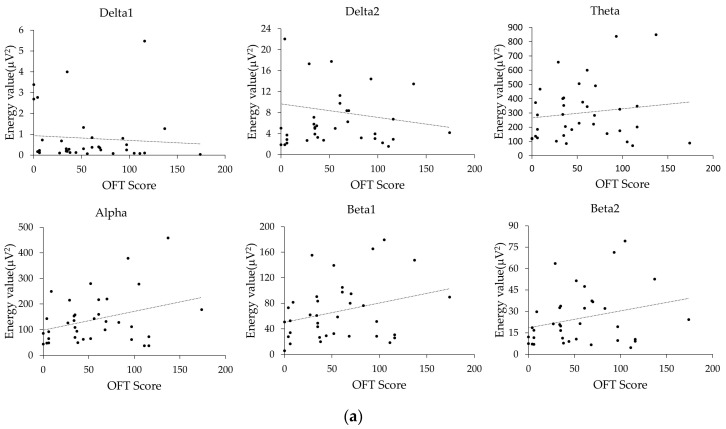
Relationship between EEG feature values and motor function recovery. (**a**) Energy in each band; (**b**) Nonlinear features; (**c**) SampEn in the sham group.

**Table 1 entropy-21-00698-t001:** Correlation between various characteristic parameters of electroencephalogram (EEG) and open field test (OFT) scores. LZC—Lempel–Ziv complexity. SampEn—sample entropy; ApEn—approximate entropy. (* indicates *p* < 0.05).

Features of EEG Signal	Correlation Coefficient (r)	*p*-Value
Delta1	−0.184	0.283
Delta2	−0. 07	0.685
Theta	0.073	0.673
Alpha	0.217	0.204
Beta1	0.209	0.221
Beta2	0.217	0.203
LZC	−0.006	0.972
ApEn	0.216	0.206
SampEn	0.393	0.018 *
ApEn (sham group)	−0.139	0.42
SampEn (sham group)	−0.025	0.886

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
