# Peer review of "Increased Sample Entropy in EEGs During the Functional Rehabilitation of an Injured Brain"

_entropy, 2019, doi:10.3390/e21070698_

Round 1

Reviewer 1 Report

I thank the authors for answering each of the observations in an appropriate manner.

Author Response

Thank you very much for your comment.

Reviewer 2 Report

This manuscript (513453) was a revision of the previously submitted one (436675) by addressing the concern of insufficient group samples. This revision performed the signal analysis and statistical assessment on a sample size of 10 or 12 for the three groups. The results now look convincing. However, few minor issues need further elaboration before publication.

1.       The quality and legend of each figure should be improved. They are now quite difficult to appreciate. In addition, the annotated mark between comparative conditions should appear only for statistical significance, which helps to improve legibility.

2.       Since the sample sizes are all below 15, a non-parametric statistical assessment should be applied instead. The small sample size may violate the assumption of data samples with a normal distribution posed in the parametric manner such as t-test and ANOVA.

3.       It seems that the manuscript did not address the OFT difference between three groups?

4.       The reviewer suggests to remove “pilot study” from the title. This work now provides informative insights of rehabilitation process in the brain upon acceptable sample sizes.

Author Response

Thank you very much for your comments.

In the revised manuscript, we have performed additional data analyses and added new figures to adequately address all concerns raised by the reviewers. The substantially revised or newly added text is highlighted by green font. For modified and new charts, the icon is also highlighted by green font. The following is a point-by-point list of all changes made in response to the comments (see attached). All page and figure numbers referred to are those of the revised manuscript.

Reviewer 3 Report

The paper is technically sound. The coverage of the topic is sufficiently comprehensive and balanced. The technical depth of the paper is appropriate for the generally knowledgeable individual Working in the Field. Anyway, the results shown in the paper are OK.

This paper is interesting the EEG community.

This is my advices to the authors:

In the abstract the authors should tell the news made in the paper. What is presented in literature?

The author written Line 65 written <<Many studies have used ApEn, SampEn and LZC to study the EEG signals of cerebral ischemia….>>

On Lines 68-69 <<In this paper, energy, ApEn, SampEn and LZC were selected to analyze the dynamic changes in EEG signals in rats with brain injury and to explore whether the features of EEG reflect neural remodeling during the recovery period.>>

On line 161 <<In this study, ApEn [25], SampEn [26] and LZC [27] were used to analyze the dynamic changes  in EEG in rats with focal brain injury.>>

What is  EEG Energy? The authors written on line 151  <<We use wavelet decomposition  to divide the signal into different frequency bands and calculate the wavelet energy. >>…on 153 <<The energy value of a certain wavelet coefficient can be used to represent the energy value of a signal in a band.>>

Energy is input  used to ApEn [25], SampEn [26] and LZC . Is it correct? On line 60 <<In this work, EEG energy values were analyzed to demonstrate changes in activity during the rehabilitation of brain injury.>>

This should be said in abstract.

In abstract should be  write that Rats in the open field test (OFT) experiment were divided into three 113 groups: an injury group (n=12), a sham group (n=12) and a control group (n=10)

Are overlying the Four time-windows of 10 s?

What is independent-sample t-test ANOVA?

I would write it.

I think a block diagram would be needed to present the steps that the authors will make in the article.

Very good is the Discussion.

The References are significant and sufficient.

Author Response

(The authors gave the same response as above.)

Reviewer 4 Report

This work presents a correlation analysis among different non-linear features and EEG signals for obtaining a rehabilitation measure after brain injury generated by ischemia. The study was led using three groups of rats (i.e., injury, control, and sham). The results demonstrated that Approximate entropy and Sample entropy as promising measures of rehabilitation level after injury by ischemia.

The work was well conducted.

Author Response

Thank you very much for your comment.

Round 2

Reviewer 3 Report

The paper is technically sound. The coverage of the topic is sufficiently comprehensive and balanced. The technical depth of the paper is appropriate for the generally knowledgeable individual Working in the Field.

In the abstract the authors show the goal of the paper.  The title is too long.  English language and style are clear.

Figures, block diagram and tables have sufficient resolution. They show the results clearly.

The results are properly commented. The bibliography is enough.

The authors have responded well to the auditor's questions and They have agreed the suggested changes.
Line 21 <<The open field test (OFT) experiment in three three groups: an injury group, a sham group and a control group (n=10) were performed.>>

This manuscript is a resubmission of an earlier submission. The following is a list of the peer review reports and author responses from that submission.

Round 1

Reviewer 1 Report

This work exploited the efficacy of EEG spectral energy and entropy-basis neuromarkers to objectively reflect the recovery of motor capacity in the injured brain. The analytical framework was empirically demonstrated on ischemia-controlled rats versus control groups. This manuscript is well organized and drafted, and the conceived signal analytical strategies did definitely help to clarify the addressed issue. However, the current form led to two major concerns regarding the insufficient number of subjects involved in the analysis and the EEG recording absent in the control group, which considerably weakened its integrity and contribution.

One major concern is that this work validated the proposed analytical framework upon the brain signals collected from six injury rats versus seven control rats only. Such an insufficient number of samples plausibly violated a basic assumption for ANOVA and T-test analysis adopted in the current work. That is, the sample distributions among groups have to comply with a normal distribution. The current statistical assessment is thus problematic.  

Another major concern is about the EEG recording absent in the control group. Their EEG oscillations along days can provide an objective benchmark of the employed spectral energy and entropy indices for the injured brain during recovery. The reason is that the amplitudes of the adopted quantitative indices tended to increase over the injured as well as non-injured brain regions along days (c.f., Fig. 2). This leads to wondering if these results were confounded by the degree of motion artifacts due to the improvement in motor capacity. Thus, the current form falls short to exclude the above concern without the EEG outcomes of the control group.

This work requires to fully disclose how to select four 10-s EEG signals from the recorded raw data for the sequential analysis. Is there any objective criterion to follow for each subject? It is also unknown how this work dealt with the notorious motion artifacts during the EEG recording.

Author Response

We thank you for your valuable comments! In the revised manuscript, we have performed additional data analyses and added new figures to adequately address all concerns raised by the two reviewers. The substantially revised or newly added text is highlighted by red font. For modified and new charts, the icon is also highlighted by red font. The following is a point-by-point list of all changes made in response to the comments. All page and figure numbers referred to are those of the revised manuscript.

Reviewer 2 Report

The work is interesting and promising, however I have some observations:

- Why were EEG records performed only in the injury group? How were the rats in the control group used?

- The paragraph after Eq. 1 is not understood, specifically when j-th layer of the EEG signal is mentioned.

- It is necessary to incorporate the equations that define ApEn, SampEn and LZC.

- Although variations were found in the proposed indexes to assess brain damage, there is no recognizable pattern of their evolution in the recovery process and therefore it is not clear how these indices would be used to evaluate the rehabilitation process of a person with brain damage.

Author Response

(The authors gave the same response as above.)

Round 2

Reviewer 1 Report

Thank the authors for responding to each of the raised issues and revising the manuscript accordingly. Especially, this revision additionally provided the comparative results from a sham group of six subjects while EEG signals were recorded. Unfortunately, the samples of each experimental group for the analytical results were all below 7 and remained insufficient for deriving scientific evidence with statistical power. The authors are strongly suggested to increase the samples to complete this study. 

Author Response

In the revised manuscript, we have performed additional data analyses  to adequately address all concerns raised by the reviewer. The substantially revised or newly added text is highlighted by blue font. For modified figures and table, the icon are also highlighted by blue font. The following is a point-by-point list of all changes made in response to the comments. All page and figure numbers referred to are those of the revised manuscript.

Reviewer 2 Report

- The paragraph after Eq. 1 is not understood, specifically when j-th layer of the EEG signal is mentioned.

The authors´s response is not correct, since according to what was  described in author_response.doxc file as well as in the modified article. A multiresolution decomposition was performed using discrete dyadic wavelet trnasform (DDWT). Consequently, the coefficients d are wavelets coefficients, not wavelets packets coefficients.

Author Response

Thank you for your careful  and found the mistake last reply to you, I am very sorry about this, I will pay more attention in the future.

Round 3

Reviewer 1 Report

This revision added the data of three and one subjects to the injury group and sham group, respectively, and re-conducted the statistical assessment using t-test and ANOVA. Unfortunately, the number of subjects recruited in each group did not substantially increase. Their sample size below 10 is still insufficient to conduct a parametric statistical analysis, which had been raised in the first-run review. The same concern remains in this revision.

Reviewer 2 Report

The observations made in the previous reviews have been satisfactorily answered.